# Associations of Lipids and Lipid-Lowering Drugs with Risk of Vascular Dementia: A Mendelian Randomization Study

**DOI:** 10.3390/nu15010069

**Published:** 2022-12-23

**Authors:** Xiaoyu Zhang, Tao Geng, Ning Li, Lijuan Wu, Youxin Wang, Deqiang Zheng, Bo Guo, Baoguo Wang

**Affiliations:** 1Department of Anesthesiology, Sanbo Brain Hospital, Capital Medical University, Beijing 100093, China; 2Geriatric Department, Emergency General Hospital, Beijing 100028, China; 3Department of Epidemiology and Health Statistics, School of Public Health, Capital Medical University, Beijing 100069, China; 4Department of Hematology, The Second Medical Centre & National Clinical Research Center for Geriatric Diseases, Chinese PLA General Hospital, Beijing 100853, China

**Keywords:** lipid-related traits, lipid-lowering drugs, vascular dementia, eQTL, Mendelian randomization

## Abstract

Accumulating observational studies suggested that hypercholesterolemia is associated with vascular dementia (VaD); however, the causality between them remains unclear. Hence, the aim of this study is to infer causal associations of circulating lipid-related traits [including high-density lipoprotein cholesterol (HDL-C), low-density lipoprotein cholesterol (LDL-C), triglyceride (TG), apolipoprotein A-I (apoA-I), and apolipoprotein B (apoB)] with VaD jointly using univariable MR (uvMR), multivariable MR (mvMR) and bidirectional two-sample MR methods. Then, the summary-data-based MR (SMR) and two-sample MR analysis were conducted to investigate the association of lipid-lowering drugs target genes expression (including *HMGCR*, *PCSK9*, *NPC1L1*, and *APOB*) and LDL-C level mediated by these target genes with VaD. The results of forward MR analyses found that genetically predicted HDL-C, LDL-C, TG, apoA-I, and apoB concentrations were not significantly associated with the risk of VaD (all *p* > 0.05). Notably, there was suggestive evidence for a causal effect of genetically predicted VaD on HDL-C via reverse MR analysis [odds ratio (OR), 0.997; 95% confidence interval (CI), 0.994–0.999; *p* = 0.022]. On the contrary, the MR results showed no significant relationship between VaD with LDL-C, TG, apoA-I, and apoB. The results for the SMR method found that there was no evidence of association for expression of *HMGCR*, *PCSK9*, *NPC1L1*, and *APOB* gene with risk of VaD. Furthermore, the result of MR analysis provided evidence for the decreased LDL-C level mediated by gene *HMGCR* reduced the risk of VaD (OR, 18.381; 95% CI, 2.092–161.474; *p =* 0.009). Oppositely, none of the IVW methods indicated any causal effects for the other three genes. Using genetic data, this study provides evidence that the VaD risk may cause a reduction of HDL-C level. Additionally, the finding supports the hypothesis that lowering LDL-C levels using statins may be an effective prevention strategy for VaD risk, which requires clinical trials to confirm this result in the future.

## 1. Introduction

Vascular dementia (VaD) is a complex neurological disease that affects memory and cognitive abilities, accounting for at least 20% of dementia worldwide [1]. VaD is generally characterized by impaired blood flow to the brain and damage to the blood vessels [2]. Accumulating evidence has suggested that vascular risk factors such as hypercholesterolemia may make a valuable contribution to VaD [3,4]. However, the associations between lipids and cognition, as well as the underlying mechanism, are complex and still unclear.

Elevated cholesterol levels are a well-established risk factor for VaD [5]. Elevated levels of low-density lipoprotein cholesterol (LDL-C) are particularly atherogenic and may be closely related to VaD [6]. In either cross-sectional or prospective analyses, higher level of LDL-C and decreased level of high-density lipoprotein cholesterol (HDL-C) was risk factors for VaD [7]. With desirable cholesterol level (<200 mg/dL) as a reference, the hazard ratio (HR) and 95% confidence interval (95% CI) for VaD was 1.50 (1.01–2.23) for borderline and 1.26 (0.82–1.96) for midlife serum high cholesterol in the cohort study [8]. Raffaitin et al. also found that a high triglyceride (TG) level was significantly associated with the incidence of VaD in a large cohort study [9]. Both TG and cholesterol are carried in plasma by apolipoprotein B (apoB)-containing lipoprotein particles. In addition, as the major component of HDL-C, apolipoprotein A-I (apoA-I) plays a critical role in reverse cholesterol transport [10]. A Swedish study found that higher apoB level at baseline predicted all forms of dementia [11]. In contrast, high levels of TC, HDL-C and TG in late life were not associated with an increased risk of VaD in some cohort studies [12]. The results in these conventional observational studies are inconsistent and may be determined to be caused by residual confounding factors and reverse causality.

As possible etiological factors of VaD are atherosclerotic vascular lesions, one of the important areas of treatment is lipid metabolism analysis and drug treatment for dyslipidemia [13,14]. Statins are regarded as the most common lipid-lowering drugs and are 3-hydroxy-3-methylglutaryl-coenzyme A reductase [HMG-CoA reductase (HMGCR)] inhibitors that are the rate-limiting enzymes in the mevalonate pathway [15]. Evolocumab or alirocumab can inhibit the proprotein convertase subtilisin-kexin type 9 (PCSK9) with monoclonal antibodies to lower LDL-C level [16]. Ezetimibe appears to act by targeting Niemann-Pick C1-like protein (NPC1L1), which is involved in cholesterol cellular uptake [17]. Additionally, as a second-generation antisense oligonucleotide, mipomersen inhibits the synthesis of apolipoprotein B-100 (APOB), which is an essential component of LDL, and thus decreases the production of LDL [18].

Similar to genetic epidemiology, the expression and function of protein drug targets can be affected by variants within or near the genes that encode them, and genetic effects can be used to anticipate the effects of lipid-lowering drug action [19]. Mendelian randomization (MR) is an alternative approach that can estimate the causal effects of exposure on the risk of diseases [20]. According to Mendel’s law, genetic material is randomly allocated at meiosis and fixed from parents to offspring at conception. Therefore, MR studies are less prone to be influenced by potential confounders and reverse causation [21]. Moreover, multivariable MR (mvMR), an extension of the inverse-variance weighted approach using genetic variants associated with one or more lipid-related traits as instrumental variables (IVs), can estimate the causal influence of each risk factor on the outcome [22].

In this study, we applied the MR approach to assessing the potential causal relationships of lipid-related traits with the risk of VaD using publicly available genome-wide association study (GWAS) data in European populations. Furthermore, we used two-sample MR analysis to test the hypothesis that a low LDL-C level due to genetic variation in *HMGCR*, *PCSK9*, *NPC1L1*, and *APOB* is associated with a risk reduction of VaD.

## 2. Materials and Methods

### 2.1. Study Design

First, two-sample univariable MR (uvMR) analyses were used to estimate the causal effects of circulating lipid-related traits on VaD using genetically predicted HDL-C, LDL-C, TG, apoA-I, and apoB levels as exposures and risk of VaD as outcomes based on summary-level GWAS datasets. Second, to assess reverse causation, bidirectional two-sample univariable MR analyses were performed to evaluate the effect of genetically predicted risk of VaD as exposures on HDL-C, LDL-C, TG, apoA-I, and apoB levels as outcomes. Third, we conducted multivariable MR (mvMR) analyses to assess the independent influence of genetically predicted HDL-C, LDL-C, TG, apoA-I, and apoB levels on the risk of VaD. Fourth, we conducted summary-data-based MR (SMR) analysis to investigate the association of gene expression from eQTL studies, in which the most significant cis-eQTL SNP was selected as a genetic instrument for the target gene expression (*HMGCR*, *PCSK9*, *NPC1L1*, and *APOB*) of each approved lipid-lowering drug, with VaD using summary data from GWAS. Finally, we used genetic variants related to LDL-C mediated by these target genes as instruments to proxy the exposure of lipid-lowering drugs and applied a two-sample MR method to explore the association between lipid-lowering drugs and VaD risk from 2 GWASs. The reporting guidelines follow the STROBE-MR statement [23]. Because this MR study was performed based on publicly available summary statistics from relevant GWASs, the ethical approval included can be found in the original articles.

### 2.2. Data Sources and Identifying Genetic Instruments

Detailed information on the summarized data sources for the instrumental variables is presented in Appendix A.

#### 2.2.1. GWAS of VaD

Summarized data on VaD were obtained from the GWAS of the FinnGen biobank with 1118 cases and 251,154 controls of European ancestry from the OpenGWAS database (https://www.finngen.fi/en/access_results (accessed on 24 January 2022)) [24].

#### 2.2.2. GWAS of Lipid-Related Traits

Summary statistical data for LDL-C (*n* = 440,546), HDL-C (*n* = 403,943), TG (*n* = 441,016), and apoA-I (*n* = 393,193) and apoB (*n* = 439,214) were extracted in a meta-analysis of GWAS involving individuals of European ancestry in the UK Biobank (UKB) [25].

All genome-wide significant genetic variants (*p* < 5 × 10^−8^) were selected as IVs. We identified independent genetic variants using the cutoff of the corresponding linkage disequilibrium (LD) value (threshold set at r^2^ < 0.001, kb = 10,000) to ensure that the IVs were independent (Appendix A) [26]. Then, we selected independent SNPs as IVs (r^2^ < 0.001) that were associated with LDL-C, HDL-C, TG, apoA-I and apoB at *p* < 5 × 10^−8^ for mvMR analysis (Appendix A). We did not include proxy SNPs and excluded palindromic SNPs with intermediate allele frequencies [26]. For each selected IV, the mean F-statistic was calculated to evaluate the strength with the approximation method described previously [27,28]. An F-statistic below 10 was considered a weak IV [29,30].

#### 2.2.3. eQTL Data

As shown in Appendix A, SNPs in 4 gene regions (*HMGCR*, *PCSK9*, *NPC1L1* and *APOB*) were used to proxy the effect of lipid-lowering drugs from available eQTLs. The *cis*-eQTL summary-level data for gene expression of *HMGCR* in blood were obtained from the eQTLGen Consortium (https://www.eqtlgen.org/ (accessed on 23 December 2019)). Additionally, the summary data for gene expression of *PCSK9* in blood, *NPC1L1* and *APOB* in subcutaneous adipose tissue were selected from GTEx Consortium V8 (https://gtexportal.org/ (accessed on 22 January 2018)). We selected *cis*-eQTL genetic instruments significantly [minor allele frequency (MAF) > 1% and *p* < 5.0 × 10^−8^] associated with the expression of genes within 1 Mb on either side of the encoded gene.

The lowering of LDL-C in circulation is an established physiological response produced by the use of lipid-lowering drugs. Hence, we identified IVs by selecting significant SNPs (*p* < 5 × 10^−8^) within 200 kb windows associated with LDL-C level at these four genomic regions from the UKB. To maximize the strength of instrumental variables for each drug, the more relaxed threshold of clumping SNPs for independence was used (r^2^ < 0.30). Then, we checked the location for the *HMGCR*, *NPCIL1*, and *APOB* genes in NCBI (https://www.ncbi.nlm.nih.gov/genome/ (accessed on 1 June 2022)) or the PCSK9 gene in the Pheno Scanner GWAS database (version 2; http://phenoscanner.medschl.cam.ac.uk (accessed on 1 June 2022)). The details of IVs in this MR analysis are presented in Appendix A.

### 2.3. MR Analyses

#### 2.3.1. MR Estimates Using uvMR

We applied a multiplicative random effects inverse-variance weighted (IVW) model as the main statistical method [31] to combine effect estimates when using genetic variants associated with the exposures as an instrument (for analysis with ≥3 SNPs); otherwise, a fixed-effects model was used. The causal estimates were obtained from a meta-analysis of SNP-specific Wald ratio estimates, which essentially translates to a weighted regression of SNP-outcome effects on SNP-exposure effects (intercept term set to zero) [29].

In sensitivity analyses, we also used other statistical methods robust to causal estimates. Specifically, weighted median (WM), penalized weighted median (PWM), MR-Egger, and Mendelian randomization pleiotropy residual sum and outlier (MR-PRESSO). The weighted median is robust to invalid instruments and could generate consistent estimates even when >50% of selected genetic variants are invalid instruments [32]. We also performed the MR-PRESSO approach to detect and correct horizontal pleiotropic outliers for all reported results in multi-instrument summary-level MR testing [33]. MR-Egger regression, which allows the intercept to be freely estimated as an indicator of average pleiotropic bias, can detect some violations of the standard instrumental variable assumptions, i.e., the InSIDE (InstrumentStrength Independent of Direct Effect) assumption is satisfied [34]. In addition, a *p*-value for the MR-Egger intercept greater than 0.05 indicates no horizontal pleiotropic effects. Cochran’s Q test was used to assess heterogeneous effects among IVs based on the IVW method [32]. The leave-one-out analysis was used to evaluate whether the association was driven by a single SNP [35].

#### 2.3.2. MR Estimates Using mvMR

We obtained the mvMR estimate by using multivariable weighted linear regression, an extension of the inverse-variance weighted method [22]. In addition, mvMR-Egger as a sensitivity analysis was performed to provide robustness against both measured and unmeasured pleiotropy [36]. To adjust pleiotropic effects across lipid traits, we performed 3 models in mvMR analyses: (1) Model 1 included HDL-C and LDL-C; (2) Model 2 included HDL-C and apoA-I; and (3) Model 3 included HDL-C, LDL-C, TG, apoA-I, and apoB.

All data analyses for MR were conducted by “MendelianRandomization” (Version 0.5.1), “TwoSampleMR” package (Version 0.5.6) [26] and “MR-PRESSO” package (Version 1.0) in the R environment (R version 4.1.2, R Project for Statistical Computing).

#### 2.3.3. SMR Analyses

We performed SMR to identify associations between gene expression and complex traits using summary data from GWAS and eQTL studies. We also performed the heterogeneity in dependent instruments (HEIDI) test to evaluate the existence of linkage in the observed association. A *P*_HEIDI_ of < 0.05 is considered evidence to support that the observed association could be due to two distinct genetic variants in high linkage disequilibrium with each other. The analyses were conducted in the SMR software tool (version 1.03, https://cnsgenomics.com/software/smr/#Overview (accessed on 28 March 2016)) [37].

A two-sided *p* value below 0.05 was considered statistically significant.

## 3. Results

### 3.1. uvMR Analysis of Lipid-Related Traits on VaD Risks via Forward MR

In uvMR, we used 280 genome-wide significant SNPs for HDL-C, 138 SNPs for LDL-C, 241 SNPs for TG, 235 SNPs for apoA-I, and 154 SNPs for apoB as IVs. All SNPs had a mean F-statistic > 10 (Table 1 and Appendix A).

The IVW estimate showed that genetically predicted HDL-C (OR, 0.852; 95% CI, 0.623–1.166; *p* = 0.317), LDL-C (OR, 0.954; 95% CI, 0.700–1.301; *p =* 0.768), TG (OR, 1.254; 95% CI, 0.950–1.656; *p* = 0.110), apoA-I (OR, 0.721; 95% CI, 0.516–1.007; *p =* 0.055), and apoB (OR, 1.213; 95% CI, 0.930–1.581; *p* = 0.154) concentrations were not significantly associated with the risk of VaD. The null associations were robust to using different analytic methods apart from the result of MR-Egger for apoA-I (*p* = 0.028) and apoB (*p* = 0.009). Furthermore, except for apoB (intercept = −0.016, *p =* 0.018), there were no directional pleiotropies for the sensitivity analyses (Table 1). Furthermore, the heterogeneities for HDL-C (*p =* 3.570 × 10^−8^), TG (*p =* 0.0003), and apoA-I (*p =* 5.845 × 10^−9^) were detected. This raised concerns for potential pleiotropic effects, and the results were further evaluated in the mvMR analysis. The scatter plots are presented in Appendix A. The leave-one-out method indicated that no SNP was substantially driving the association between lipid-related traits and VaD risks (Appendix A).

### 3.2. Causal Effects of VaD on Lipid-Related Traits via Reverse MR Analyses

As shown in Figure 1 and Appendix A, the IVW method estimate using 5 SNPs showed that there was suggestive evidence for a causal effect of genetically predicted VaD on HDL-C (OR, 0.997; 95% CI, 0.994–0.999; *p* = 0.022), with no evidence of heterogeneity (*p* = 0.977). In contrast, the IVW MR results showed no significant relationship of VaD with LDL-C (IVs, 5; OR, 1.002; 95% CI, 0.998–1.005; *p* = 0.327), TG (IVs, 5; OR, 1.001; 95% CI, 0.998–1.004; *p* = 0.380), apoA-I (IVs, 5; OR, 0.998; 95% CI, 0.995–1.001; *p* = 0.107), or apoB (IVs, 5; OR, 1.002; 95% CI, 0.999–1.005; *p =* 0.226). Similar results were presented in four other MR methods. Likewise, no heterogeneity or directional pleiotropy was detected based on the Q test (all *p* >0.05) and MR-Egger intercept test (all *p* > 0.05). The associations between the effect sizes of the SNP–VaD relationship and the SNP–the phenotypes of lipid-related trait relationships are presented in Figure 2 and Appendix A. The results of the leave-one-out sensitivity analysis found that the association between genetically determined VaD and lipid-related traits was not significantly changed (Figure 3 and Appendix A).

### 3.3. mvMR Analysis of Lipid-Related Traits in VaD Risk

To control for pleiotropic pathways, we estimated the direct effect of each exposure in three models performing mvMR analysis (Appendix A). In model 1, there was a nonsignificant association for HDL-C (IVs, 328; OR, 1.023; 95% CI, 0.835–1.255; *p* = 0.824) and LDL-C (IVs, 328; OR, 0.918; 95% CI, 0.730–1.154; *p* = 0.462) with the risk of VaD based on multivariable IVW MR analyses. In model 2, the results from mvMR analyses found no evidence for the associations of HDL-C (IVs, 323; OR, 0.766; 95% CI, 0.384–1.528; *p* = 0.450) and apoA-I (IVs, 323; OR, 1.148; 95% CI, 0.537–2.456; *p* = 0.722) with the risk of VaD. In model 3, the associations of HDL-C (IVs, 416; OR, 0.634; 95% CI, 0.214–1.879, *p* = 0.412), LDL-C (IVs, 416; OR, 1.212; 95% CI, 0.441–3.331, *p* = 0.710), TG (IVs, 416; OR, 0.869; 95% CI, 0.606–1.247, *p* = 0.449), apoA-I (IVs, 416; OR, 1.449; 95% CI, 0.549–3.824, *p* = 0.454), and apoB (IVs, 416; OR, 0.863; 95% CI, 0.374–1.994; *p* = 0.730) with the risk of VaD were also nonsignificant. However, except for model 1 (mvMR-Egger: *p* = 0.041; Q statistic: *p* < 0.001), Cochrane’s Q statistic and mvMR-Egger test did not detect heterogeneity and pleiotropy. These results were consistent with the uvMR analyses.

### 3.4. SMR Analyses

The SMR method was conducted to assess the association between the expression of *HMGCR*, *PCSK9*, *NPC1L1*, and *APOB* and VaD outcome. We obtained 921, 24, 11, and 161 SNPs for *cis*-eQTL results from eQTLGen or GTEx Consortium about the drug target genes *HMGCR*, *PCSK9*, *NPC1L1*, and *APOB*, respectively. Next, SMR analyses were performed to use the most significant *cis*-eQTL SNP (rs6453133, rs472495, rs41279633, and rs4665179) as an IV for the target gene of each lipid-lowering drug (Appendix A).

There was no evidence of a significant association between the expression of *HMGCR* (*β*, −0.458; *p* = 0.264), *PCSK9* (*β*, −0.091; *p* = 0.732), *NPC1L1* (*β*, −0.078; *p* = 0.547), and *APOB* (*β*, 0.010; *p* = 0.937) genes and the risk of VaD. The HEIDI test suggested that all observed associations were not due to a linkage (*p* > 0.05), except for *NPC1L1* (*p =* 0.005) expression.

### 3.5. Causal Effect of LDL-C Level Mediated by Target Genes on VaD via MR Analyses

We selected 7, 30, 6, and 39 SNPs within or near the genes *HMGCR*, *PCSK9*, *NPC1L1*, and *APOB* from summary statistical data for LDL-C level from the UKB, respectively (Table 2 and Appendix A).

The results of MR analysis provided evidence that the decreased LDL-C level mediated by the *HMGCR* gene reduced the risk of VaD (OR, 18.381; 95% CI, 2.092–161.474; *p* = 0.009), suggesting little evidence of heterogeneity (*p* = 0.916). In contrast, none of the IVW MR methods indicated any causal effects of genetically predicted LDL-C levels mediated by the *PCSK9* (OR, 0.783; 95% CI, 0.466–1.316; *p* = 0.356), *NPC1L1* (OR, 0.139; 95% CI, 0.019–1.017; *p* = 0.052), and *APOB* (OR, 1.318; 95% CI, 0.767–2.264; *p* = 0.318) genes on the risk of VaD.

A lack of causal association remained in all sensitivity analyses (all *p* > 0.05), except for the *NPC1L1* gene based on MR-PRESSO analysis (*p* = 0.049). Likewise, there was no clear evidence of heterogeneity (all *p* > 0.05) or pleiotropy (all *p* values for intercept > 0.05). Appendix A show the individual MR estimates of the causal effect of LDL-C levels mediated by target genes on the risk of VaD in GWAS datasets based on the IVW method. Leave-one-out sensitivity analyses did not detect any significant changes (Appendix A).

## 4. Discussion

Using an integrated approach, including conventional uvMR, mvMR and SMR analyses, our study aimed to assess the causal effects of genetically determined lipid-related traits and lipid-lowering drugs on the risk of VaD in a European population. Our study found that there was suggestive evidence for a causal effect of genetically determined VaD on HDL-C level, and the decreased LDL-C level mediated by the *HMGCR* gene could reduce the risk of VaD.

The current literature has examined the associations between lipid concentrations and incident VaD risk, but no definitive conclusions can be drawn. Several studies reported no significant association between higher TC concentration with an increased risk of VaD [38,39]. There was a relationship between LDL-C level and the risk of VaD in one study [7], whereas many studies reported no significant associations with VaD risk [12]. In addition, there was no significant association between TG concentration level and risk of VaD in the vast majority of studies [9,40,41]. Likewise, many studies have reported no significant association between lower HDL-C concentration with an increased risk of VaD risk [9,38,40,41]. The Copenhagen General Population Study and Copenhagen City Heart Study also found that the hazard ratio (HR) for a 1 mmol/L lower observational LDL-C level was 1.09 (0.97 to −1.23) for VaD [42]. Thus, it is possible that these inconsistent findings resulted from confounding by unmeasured/poorly measured confounders and reverse causation in observational studies. Our uvMR and mvMR analyses also support no evidence that lipid-related traits have an effect on the risk of developing VaD from European descent. The mechanisms underlying the association between lipid fractions and VaD risk may be directly mediated by cerebrovascular disease.

The relationship between plasma lipids and cognition is very complex and controversial. Cholesterol is a major constituent of the myelin-encircling neurons in the brain and the risk of neurological diseases [43]. To date, many studies have not drawn any conclusions on the impact of lipid-lowering drugs, specifically statins, on VaD risk. Categories of lower LDL-C levels reflecting values recommended for lipid-lowering treatment, with more than 97% accounted for by statins (<1.8 mmol/L, 1.8–2.59 mmol/L, 2.6–3.99 mmol/L, and ≥4 mmol/L), were not associated with the risk of VaD (*p* for trend = 0.560) in a cohort study [42]. Statins may slow the rate of cognitive decline and delay the onset of all-cause dementia in cognitively healthy elderly individuals [44]. The prospective observational associations of lipid-lowering drug use with VaD risk would also be prone to bias via residual confounding by indication.

This study found that there was suggestive evidence for a causal effect of genetically predicted VaD on HDL-C level. Dementia is a progressive and largely irreversible clinical syndrome, including mental function impairment, characterized by memory, language, activities of daily living, and psychosocial and psychiatric disturbance [45]. It is difficult to maintain a healthy lifestyle after dementia, such as physical activity, so HDL-C level may decrease. In addition, VaD is caused by different vascular etiologies, which damage blood vessels in the brain and even decrease their ability to supply sufficient oxygen and nutrients that enable the brain to function effectively [46]. Furthermore, hypoperfusion causes blood-brain barrier (BBB) disruption, glial activation, oxidative stress, and oligodendrocyte loss [47]. Therefore, the reverse effect of HDL particles on oxidized LDL particles may be inhibited.

In our study, there was no evidence of a significant association between the expression of *HMGCR*, *PCSK9*, *NPC1L1*, and the *APOB* gene and the risk of VaD based on SMR analyses. Likewise, the results of IVW-MR analyses did not provide any evidence for the causal effects of LDL-C levels mediated by *PCSK9*, *NPC1L1*, and the *APOB* gene on the risk of VaD, except for the *HMGCR* gene. However, lowering peripheral LDL-C levels mediated by the *HMGCR* gene (the target gene of statins) has a role in decreasing VaD risk.

There are several notable strengths in our study. Unlike other studies, our study performed uvMR, mvMR and SMR analyses to evaluate the causal effects of genetically determined lipid-related traits and lipid-lowering drugs on the risk of VaD based on GWAS and eQTL data. Moreover, the SMR analytic framework was used to test for pleiotropic association/potentially causal association between the expression level of a gene and VaD using summary-level data from GWAS and expression quantitative trait loci (eQTL) studies. This design technique can avoid reverse causation and reduce other confounding factors. Finally, our results were less susceptible to population stratification bias because we confined the population for the summary data to European ancestry.

Despite its novelty, there are several limitations in this study. First, the number of VaD cases was still relatively small, which may affect the statistical power of the results in our study. Second, our data are restricted to European ancestry, and therefore, more work is required to determine if our results translate to Asian or other ancestry groups. Third, because of the unavailability of individual data, the use of summary-level data could hamper stratified analyses (e.g., coronary heart disease) or analyses adjusted for other covariates (e.g., movements, sports, income, and education). In addition, publicly genome-wide association study (GWAS) data on these lipid markers/parameters (such as LDL/HDL, oxLDL, sdLDL, etc.) in European populations were not available. However, we can explore causality in the case of a database in the future. Fourth, horizontal pleiotropy could not be excluded from our study because the results of the HEIDI test were significant for the *NPC1L1* gene. Finally, as a major regulator of lipid metabolism in the body, the liver plays a central role in the synthesis and degradation of fatty acids. Unfortunately, eQTL data in the liver is not available; hence, SMR analysis using eQTL data in the liver is expected to be performed in the future.

## 5. Conclusions

In conclusion, using genetic data, this study provides evidence that incident VaD may cause a reduction in circulating HDL-C. Additionally, the findings support the hypothesis that lowering LDL-C level using statins may be an effective prevention strategy for incident VaD.

## Figures and Tables

**Figure 1 nutrients-15-00069-f001:**
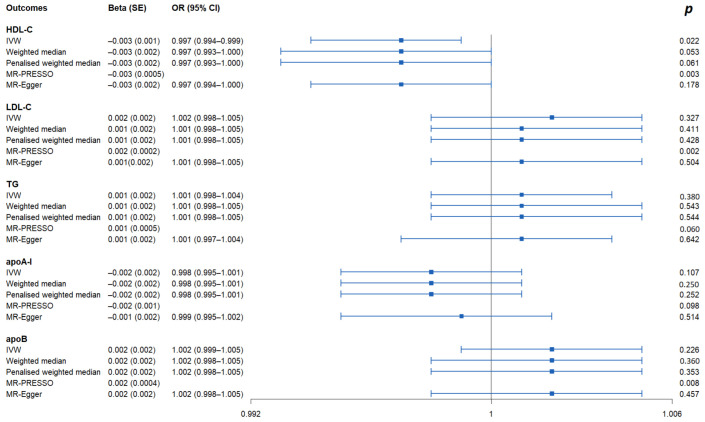
Causal effects of VaD on lipid-related traits via reverse MR analyses. IVs: instrumental variables; IVW: inverse-variance weighted; MR: Mendelian randomization; PRESSO: pleiotropy residual sum and outlier; OR: odds ratio; CI: confidence interval; HDL-C: high-density lipoprotein cholesterol; LDL-C: low-density lipoprotein cholesterol; TG: triglyceride; apoA-I: apolipoprotein A-I; apoB: apolipoprotein B.

**Figure 2 nutrients-15-00069-f002:**
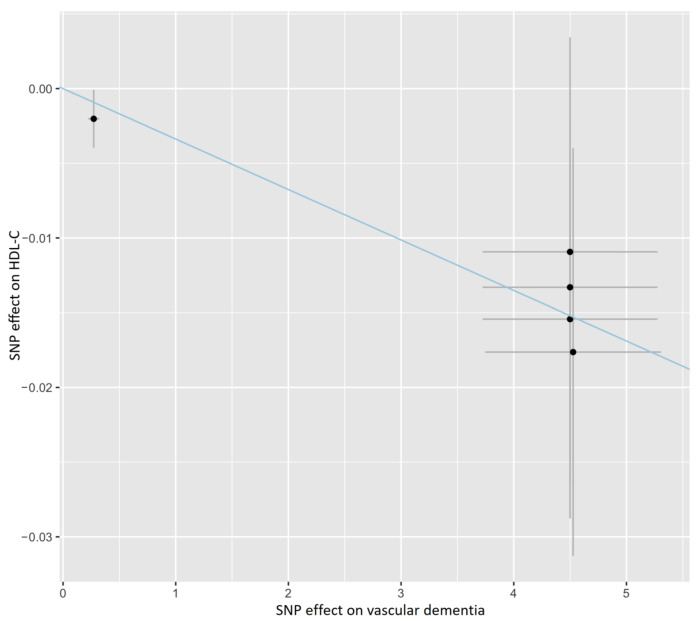
Scatter plot showing the association of the SNP effects on VaD against the SNP effects on the HDL-C level. The blue line indicates the estimate of the effect using the IVW method. Circles indicate marginal genetic associations with VaD and risk of HDL-C level for each variant. Error bars indicate 95% CIs. IVW: inverse-variance weighted; VaD: vascular dementia; HDL-C: high-density lipoprotein cholesterol; SNP: single nucleotide polymorphism.

**Figure 3 nutrients-15-00069-f003:**
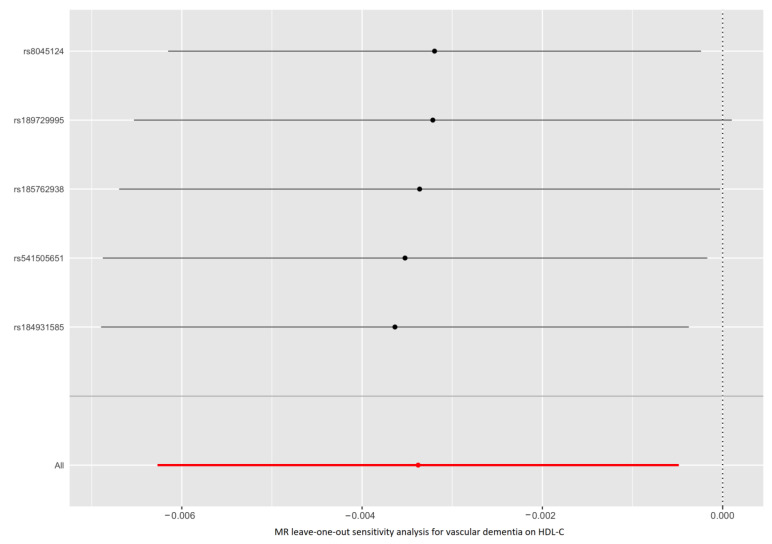
Leave-one-out permutation analysis of the causal association between VaD and HDL-C level. VaD: vascular dementia; HDL-C: high-density lipoprotein cholesterol; MR: Mendelian randomization; SNP: single nucleotide polymorphism.

**Table 1 nutrients-15-00069-t001:** Causal effects of lipid-related traits levels on vascular dementia via univariable MR analyses.

Phenotype	IVs	OR (95% CI)	Beta (SE)	*p*	Q Statistic_*p*
HDL-C					
IVW	280	0.852 (0.623–1.166)	−0.160 (0.160)	0.317	3.570 × 10^−8^
Weighted median	280	0.848 (0.543–1.324)	−0.165 (0.227)	0.468	
Penalised weighted median	280	0.859 (0.555–1.328)	−0.152 (0.222)	0.494	
MR-PRESSO (Outlier-corrected)	280		0.019 (0.124)	0.881	
global test				<0.001	
MR-Egger	280	0.774 (0.478–1.252)	−0.256 (0.245)	0.297	
egger_intercept			0.004 (0.007)	0.605	
LDL-C					
IVW	138	0.954 (0.700–1.301)	−0.047 (0.158)	0.768	0.677
Weighted median	138	0.836 (0.502–1.391)	−0.179 (0.260)	0.491	
Penalised weighted median	138	0.753 (0.455–1.245)	−0.284 (0.257)	0.269	
MR-PRESSO	138		−0.018 (0.153)	0.909	
global test				0.667	
MR-Egger	138	1.270 (0.803–2.009)	0.239 (0.234)	0.308	
egger_intercept			−0.013 (0.008)	0.100	
TG					
IVW	241	1.254 (0.950–1.656)	0.227 (0.142)	0.110	0.0003
Weighted median	241	1.143 (0.770–1.697)	0.134 (0.202)	0.506	
Penalised weighted median	241	1.140 (0.770–1.686)	0.131 (0.200)	0.513	
MR-PRESSO (Outlier-corrected)	241		0.035 (0.121)	0.770	
global test				<0.001	
MR-Egger	241	1.191 (0.794–1.786)	0.175 (0.207)	0.399	
egger_intercept			0.002(0.007)	0.730	
apoA-I					
IVW	235	0.721 (0.516–1.007)	−0.327 (0.170)	0.055	5.845 × 10^−9^
Weighted median	235	0.995 (0.630–1.571)	−0.005 (0.233)	0.981	
Penalised weighted median	235	1.042 (0.678–1.601)	0.041 (0.219)	0.852	
MR-PRESSO (Outlier-corrected)	235		−0.088 (0.134)	0.513	
global test				<0.001	
MR-Egger	235	0.549 (0.322–0.935)	−0.600 (0.272)	0.028	
egger_intercept			0.010 (0.008)	0.200	
apoB					
IVW	154	1.213 (0.930–1.581)	0.193 (0.135)	0.154	0.564
Weighted median	154	1.354 (0.847–2.164)	0.303 (0.239)	0.205	
Penalised weighted median	154	1.049 (0.682–1.615)	0.048 (0.220)	0.827	
MR-PRESSO	154		0.198 (0.134)	0.140	
global test				0.500	
MR-Egger	154	1.605 (1.130–2.281)	0.473 (0.179)	0.009	
egger_intercept			−0.016 (0.007)	0.018	

CI: confidence intervals; IVs: instrumental variables; IVW: inverse-variance weighted; MR: mendelian randomization; MR-PRESSO: Pleiotropy Residual Sum and Outlier; OR: odds ratio; SE: standard error; HDL-C: high-density lipoprotein cholesterol; LDL-C: low-density lipoprotein cholesterol; TG: triglyceride; apoA-I: apolipoprotein A-I; apoB: apolipoprotein B.

**Table 2 nutrients-15-00069-t002:** Causal effects of LDL-C mediated by target genes on vascular dementia via MR analyses.

Gene	IVs	OR (95% CI)	Beta (SE)	*p*	Q Statistic_*p*
*HMGCR*					
IVW	7	18.381 (2.092–161.474)	2.911 (1.109)	0.009	0.916
Weighted median	7	14.580 (0.949–223.977)	2.680 (1.394)	0.055	
Penalised weighted median	7	14.580 (0.967–219.879)	2.680 (1.384)	0.053	
MR-PRESSO	7		2.911 (0.646)	0.004	
global test				0.942	
MR-Egger	7	55.727 (0.112–27,641.808)	4.020 (3.167)	0.260	
egger_intercept			−0.027 (0.071)	0.724	
*PCSK9*					
IVW	30	0.783 (0.466–1.316)	−0.245 (0.265)	0.356	0.731
Weighted median	30	0.651 (0.331–1.281)	−0.429 (0.345)	0.214	
Penalised weighted median	30	0.651 (0.320–1.326)	−0.429 (0.363)	0.237	
MR-PRESSO	30		−0.245 (0.241)	0.318	
global test				0.756	
MR-Egger	30	0.679 (0.335–1.377)	−0.388 (0.361)	0.292	
egger_intercept			0.012 (0.021)	0.565	
*NPC1L1*					
IVW	6	0.139 (0.019–1.017)	−1.971 (1.014)	0.052	0.728
Weighted median	6	0.115 (0.01–1.287)	−2.160 (1.231)	0.079	
Penalised weighted median	6	0.115 (0.011–1.225)	−2.160 (1.206)	0.073	
MR-PRESSO	6		−1.971 (0.761)	0.049	
global test				0.779	
MR-Egger	6	1.239 (0.002–674.12)	0.214 (3.214)	0.950	
egger_intercept			−0.056 (0.078)	0.513	
*APOB*					
IVW	39	1.318 (0.767–2.264)	0.276 (0.276)	0.318	0.453
Weighted median	39	0.995 (0.446–2.219)	−0.005 (0.409)	0.989	
Penalised weighted median	39	0.995 (0.435–2.277)	−0.005 (0.422)	0.990	
MR-PRESSO	39		0.276 (0.276)	0.324	
global test				0.451	
MR-Egger	39	1.309 (0.427–4.010)	0.269 (0.571)	0.640	
egger_intercept			0.0003 (0.025)	0.990	

The italic for *HMGCR*, *PCSK9*, *NPC1L1*, and *APOB* indicates gene; CI: confidence interval; IVs: instrumental variables; IVW: inverse-variance weighted; MR: mendelian randomization; MR-PRESSO: Pleiotropy Residual Sum and Outlier; OR: odds ratio; SE: standard error; LDL-C: low-density lipoprotein cholesterol; *HMGCR*: 3-hydroxy-3-methylglutaryl-coenzyme A reductase; *PCSK9*: proprotein convertase subtilisin-kexin type 9; *NPC1L1*: Niemann-Pick C1-like protein; *APOB*: apolipoprotein B.

## Data Availability

The summary statistics used in the current study are available from the corresponding author upon reasonable request. Detailed information on the summarized data sources for the instrumental variables is presented in Appendix A.

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
