# Peer review of "Associations of Lipids and Lipid-Lowering Drugs with Risk of Vascular Dementia: A Mendelian Randomization Study"

_nutrients, 2022, doi:10.3390/nu15010069_

Round 1

Reviewer 1 Report

The present manuscript “Associations of lipids and lipid-lowering drugs with risk of vascular dementia: a Mendelian randomization study” is an interesting approach and very important in the context of determining factors for vascular dementia. Research in this area is highly important and the actual statistical perspective, improves the knowledge in this research area. Although, there are few additional issues that should be addressed and might further improve the quality of the paper.

I do have a manuscript version without numbered lines (to your information)

The first part of the abstract (first 2 sentences) should be rephrased  – it is not obvious to all readers that MR is used as a result of the first sentence… it should be written more clear that MR is a reliable approach for testing observational studies statistically.

Should be discussed: Do you assume that the use of statins effected the results of MR in VaD regarding LDL-C levels. Could there be an effect missing in analysis?

May it be relevant to consider other lipid marker/parameters – such as LDL/HDL subclasses, oxLDL, sdLDL…? And how might inflammation processes be involved? Discuss these issues and add information to the introduction/discussion

How are further (background) variables considered that might influence the results e.g. movements, sports, income, education? Add to discussion

Small issue – stringency: numbers should be written the same type eg. 123987 or 123,987 is mixed now

HR – abbreviation/description is missing

This part (below) needs to be rephrased – at the moment it seems more of a random assortment of phrases and it is not very soundfull, overall the discussion should be improved for its style and soundfullness. E.g. Dementia has been reported in people treated with traditional choles-terol-lowering drugs, particularly statins [44,45]. The PROspective Study of Pravastatin in the Elderly at Risk (PROSPER) trial showed no difference in global cognitive function (MMSE) after a mean follow-up period of 42 months in subjects treated with pravastat compared to placebo (all p >0.05) [46]. Statins may slow the rate of cognitive decline and delay the onset of all-cause dementia in cognitively healthy elderly individuals

Reviewer 2 Report

I read with great enthusiasm the manuscript entitled "Associations of lipids and lipid-lowering drugs with risk of vascular dementia: a Mendelian randomization study". The entire paper is beautifully and systematically written and easy to understand. Although the authors honestly and clearly point out the limitations of the study, the shortcomings are overcome by the idea and novelty of the study and the results recorded. I have no objection to this manuscript and propose to be published in current form.

Author Response

Response: Many thanks for your comment. According to your suggestion, we revised the language and grammar carefully.